# Diverse secondary metabolites are expressed in particle-associated and free-living microorganisms of the permanently anoxic Cariaco Basin

David Geller-McGrath[1,5], Paraskevi Mara [2,5], Gordon T. Taylor [3], Elizabeth Suter[3,4], Virginia Edgcomb [2,6] ✉ & Maria Pachiadaki [1,6] ✉

Secondary metabolites play essential roles in ecological interactions and nutrient acquisition, and are of interest for their potential uses in medicine and biotechnology. Genome mining for biosynthetic gene clusters (BGCs) can be used for the discovery of new compounds. Here, we use metagenomics and metatranscriptomics to analyze BGCs in free-living and particle-associated microbial communities through the stratified water column of the Cariaco Basin, Venezuela. We recovered 565 bacterial and archaeal metagenome-assembled genomes (MAGs) and identified 1154 diverse BGCs. We show that differences in water redox potential and microbial lifestyle (particle-associated vs. free-living) are associated with variations in the predicted composition and production of secondary metabolites. Our results indicate that microbes, including understudied clades such as Planctomycetota, potentially produce a wide range of secondary metabolites in these anoxic/euxinic waters.

Secondary metabolites are low-molecular-mass compounds that are not required for the growth or reproduction of an organism. Nonetheless, they can serve a variety of functions, including the facilitation of intercellular communication, inhibition of competitors, nutrient acquisition, and interactions with the surrounding environment[1]. Many classes of these molecules can have antibiotic properties, such as polyketides, non-ribosomal peptides, and ribosomally synthesized post-translationally modified peptides (RiPPs)[2]. Other examples of these compounds include terpenes, aryl polyenes, and lactones with diverse roles (e.g., pigments and quorum sensing). Groups of co-located genes, referred to as biosynthetic gene clusters (BGCs), encode instructions to build these molecules. While the chemical structures of secondary metabolites vary significantly, the biosynthetic gene sequences that encode them are often highly conserved[3]. The high similarity of the amino acid sequences of core biosynthetic enzymes facilitates the mining of genome data for the presence of specific classes of BGCs. Core biosynthetic genes are frequently flanked by regulatory, export, and resistance genes, as well as genes encoding tailoring enzymes that modify the compound scaffold[3]. Genome mining has revealed that the secondary metabolic potential of both prokaryotes and eukaryotes is much broader than what is observed under laboratory conditions[4,5]. This could be due to the absence of specific stimuli in laboratory settings that are requisite to upregulate or activate compound production in cultures[6].

Large genome mining efforts have revealed widespread and diverse biosynthetic capability among prokaryotes; yet, "extreme" environments such as oxygen-depleted water columns (ODWCs) are underrepresented in these studies[4,5]. ODWCs are oceanic realms with low (<20 μM) to undetectable oxygen concentrations[7], and include permanently-stratified basins as well as oxygen minimum zones.

[1]Biology Department, Woods Hole Oceanographic Institution, Woods Hole, MA, USA. [2]Geology & Geophysics Department, Woods Hole Oceanographic Institution, Woods Hole, MA, USA. [3]School of Marine and Atmospheric Sciences, Stony Brook University, Stony Brook, NY, USA. [4]Biology, Chemistry and Environmental Studies Department, Molloy College, Rockville Centre, NY, USA. [5]These authors contributed equally: David Geller-McGrath, Paraskevi Mara. [6]These authors jointly supervised this work: Virginia Edgcomb, Maria Pachiadaki. ✉e-mail: vedgcomb@whoi.edu; mpachiadaki@whoi.edu

ODWCs have expanded and intensified globally over the past 50 years[8] due to global climate change and anthropogenic pollution. This expansion causes changes in water column stratification and upper water column primary production, and results in shifts in the cycling of trace gases that produce feedback on climate (e.g., methane, nitrous oxide, and carbon dioxide)[9]. The Cariaco Basin is a permanently-stratified marine system off the north coast of Venezuela. The Basin's water column is fully oxic at the surface but stratification below the mixed layer (<80 m)[10,11] causes a sharp oxygen decline. A strong vertical redox gradient (redoxcline) extends from -200 m to -250–350 m depth, where oxygen becomes undetectable. Below 350 m, the water becomes euxinic with sulfide concentrations approaching 80 μM near the basin floor[12,13]. This relatively stable redoxcline makes the Cariaco Basin an ideal natural laboratory for studying how microbes organize and function in specific redox conditions. ODWCs are relatively well-studied regarding the microbially mediated biogeochemical transformations of carbon, nitrogen, sulfur, and redox-sensitive trace metals[14–17]. However, secondary metabolite genomic potential and expression in ODWCs has not yet been studied. Further, analyses of size-fractionated water samples are required in order to assess the role of particles in the production of secondary metabolites in the environment. Particles provide colonizable, nutrient-rich substrates where metabolites can be concentrated and exchanged and can provide protection for oxygen- or sulfide-sensitive microbiota.

In order to address this critical gap in secondary metabolite knowledge and assess the role of particles in the production of secondary metabolites in ODWC environments, we analyzed size-fractionated water samples along various oxygen and sulfide regimes in the water column of the Cariaco Basin. We reconstructed 565 metagenome-assembled genomes (MAGs) and we estimated their relative abundance and fraction partitioning along Cariaco's redoxcline using metagenomic read recruitment and DESeq2[18], and we identify the encoded BGCs using antiSMASH[19]. For this environmental survey of secondary metabolites, we use metatranscriptomes constructed from in situ filtration and preservation of water samples to compare the biosynthetic transcript expression profiles of particle-associated (PA >2.7 μm) and free-living (FL; 0.2-2.7 μm) fractions. In situ filtration and fixation minimize artifacts that can be introduced into RNA pools due to sample handling and physicochemical changes[20]. The detected biosynthetic clusters encode for the production of auxiliary compounds with chemical diversity and bioactivity that can provide competitive advantages via antimicrobial compounds (e.g., non-ribosomal peptide synthetases [NRPS], polyketides, RiPPs), or can have a broader impact on microbial survival via the synthesis of pigments and toxins (e.g., aryl polyenes and terpenes) or via their possible role in biofilm formation (e.g., RiPPs and phenazines).

## Results

We recovered 565 metagenome-assembled genomes (MAGs) with ≥75% bin completeness and ≤5% bin contamination from sulfidic layers of Cariaco Basin using a PA and FL size fraction co-assembly. Recovered MAGs belonged to 44 bacterial and eight archaeal phyla (Supplementary Figs. 1, 2 and Supplementary Data 1, 2). The overall taxonomic profile resembled patterns previously observed in MAGs recovered from the Black Sea[21]. Nonetheless, while identified MAGs from the Black Sea affiliated with the Bdellovibrionota and Nitrospirota phyla were not recovered in our Cariaco samples, we did recover genomes from 26 phyla not reported thus far from the Black Sea (Supplementary Data 3). This was likely due to the analysis of two different size fractions in the present dataset.

### Differential abundance (fraction partitioning) of recovered genomes

Differential abundance analysis revealed size fraction partitioning of the recovered MAGs from a taxonomic perspective, and is largely

consistent with marker gene profiles from the same samples[22]. The majority of MAGs from the Planctomycetota, Myxococcota, Verrumicrobiota, and the candidate phyla Krumholzibacteriota were differentially abundant in PA metagenomes (Supplementary Fig. 3). Planctomycetota and Verrumicrobiota were previously reported to be more abundant in the PA fraction of 16 S rRNA gene amplicon samples in various marine environments[23–27]. However, the two Planctomycetota MAGs belonging to the genus Scalindua, a group known to perform anaerobic ammonia oxidation (anammox)[28], were more abundant in the FL metagenomes as shown previously[29]. Proteobacteria (primarily *Alphaproteobacteria* and *Gammaproteobacteria*), Nanoarchaeota, Crenarchaeota, and Iainarchaeota, as well as, the candidate phyla Omnitrophota, Marinisomatota, Margulisbacteria, SAR324, and Patescibacteria were more abundant in the FL fraction. MAGs from the Desulfobacterota and Thermoplasmatota did not exhibit a preferred association with either fraction at oxycline and shallow anoxic depths, while the majority of MAGs from these phyla were more abundant in the FL fraction at the euxinic depth.

### Identification of secondary metabolite biosynthetic gene clusters

Anaerobic/microaerophilic bacteria and archaea have been overlooked as potential sources of bioactive secondary metabolites[30]. Yet, genomic studies now show that these organisms can contain enormous biosynthetic potential, much of which remains unknown[31]. The antiSMASH 6 pipeline identified and annotated 1154 BGCs longer than 10 kb (1369 total clusters identified), which contained 23,845 genes, in 68% of the recovered MAGs in this study (Supplementary Fig. 4 and Supplementary Data 4). The majority of BGCs detected in our study encoded for RiPP (332 BGCs), terpene (191 BGCs), non-ribosomal peptide (NRPS: 113 BGCs), and polyketide synthases (112 BGCs; PKS types I, II, and III). There were additionally 130 hybrid clusters composed of overlapping BGCs (e.g., non-ribosomal peptide-polyketide combinations), as well as four ectoine clusters (Supplementary Data 4). Sixty-five percent of the BGCs had a boundary on a contig edge, indicating potentially incomplete recovery of the whole sequence for nearly two-thirds of our predicted BGCs. BiG-SCAPE[32] analysis revealed that the majority of detected BGCs longer than 10 kb did not cluster together with the other 1154 biosynthetic clusters. BiG-SCAPE created 15 gene cluster families (GCFs) of size 2, while 1139 clusters were placed into singleton GCFs. Antibiotic Resistance Target Seeker[33,34] identified putative antibiotic resistance genes within some BGCs.

### Distribution and expression of secondary metabolite biosynthetic gene clusters (BGCs) across recovered MAGs

We detected BGCs in MAGs recovered from most of the recovered phyla; exceptions in BGC detection were the bacterial phyla Spirochaetota, Ratteibacteria, Dadabacteria, Dependentiae, and Aerophobota that were underrepresented (1–2 MAGs per phylum), and the archaeal phylum Iainarchaeota where ten genomes were reconstructed. The lack of detection of BGCs in the aforementioned phyla can be attributed to the small sample size analyzed.

MAGs from phyla Myxococcota, Verrucomicrobiota, and Acidobacteriota were more prevalent within the PA fraction and contained some of the largest diversity of biosynthetic gene clusters among phyla with an apparent PA preference (Fig. 1a). This likely reflects adaptations to a particle-associated lifestyle where intercellular communication and competition are relatively intense compared to the free-living state, as revealed in culture studies[35,36]. Genomes from these phyla contained several classes of BGCs, including RiPPs and NRPS, as well as polyketide and terpene synthases (Fig. 1a).

The recovered Cariaco MAGs expressed BGCs in all the PA and FL metatranscriptomes (Fig. 2). BGCs in the PA fraction were expressed

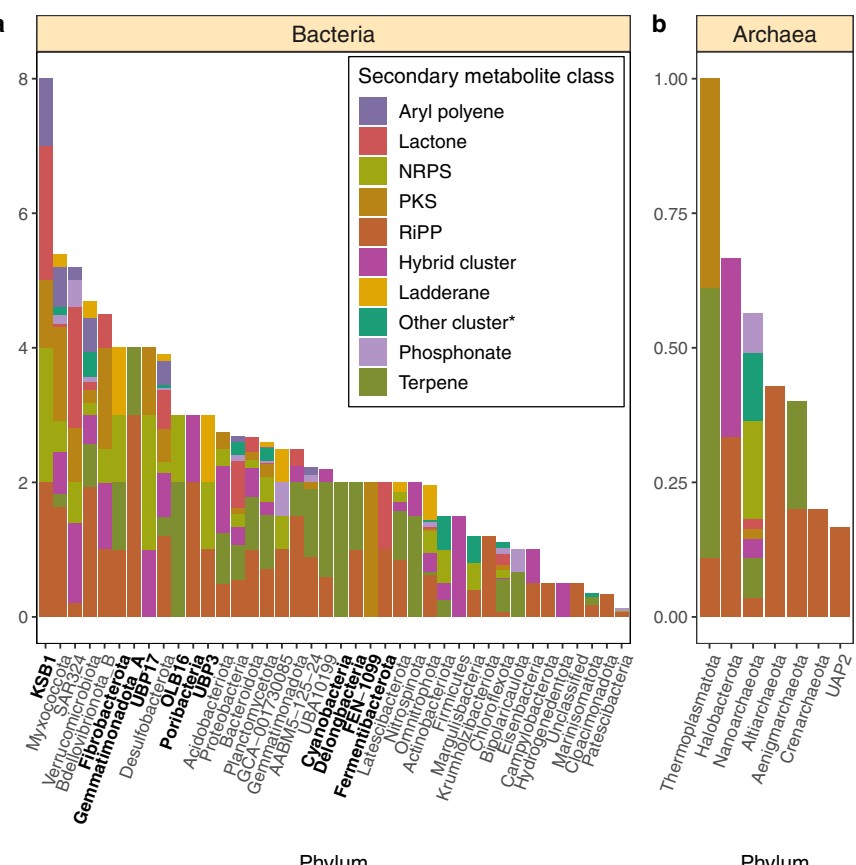

**Fig. 1 | Normalized biosynthetic gene cluster count per phylum. a** Normalized bacterial biosynthetic gene cluster count by phylum. **b** Normalized archaeal biosynthetic gene cluster count by phylum. Bold labels denote underrepresented phyla (phyla with only one representative MAG). BGC counts were normalized by dividing the total count of each BGC type present in a phylum by the total MAGs within that phylum. Source data are provided as a Source Data file.

predominantly from MAGs affiliated with the Omnitrophota, Desulfobacterota, Planctomycetota, Myxococcota, and Gammaproteobacteria. Planctomycetota genomes have recently been reported to contain diverse biosynthetic potential[37], while the Gammaproteobacteria and Myxococcota are prolific producers of secondary metabolites[30,38]. The majority of expressed BGCs in the PA fraction encoded terpenes, RiPPs, and non-ribosomal peptides. Notably, 2,980 biosynthetic genes were only expressed in PA metatranscriptomes, while 1901 were exclusively expressed in FL. Additionally, the PA fraction showed higher differential expression of 21 ($P < 0.05$; false discovery rate (FDR) = 5%) BGC genes across all sampled depths, while 138 transcripts were significantly more abundant in the FL samples. MAGs more abundant in PA metagenomes ($P < 0.05$; FDR = 5%) exhibited expression of BGC genes almost exclusively from PA metatranscriptomes with little evidence of expression in FL samples (Supplementary Fig. 5a–c). This suggests MAGs with an apparent PA preference primarily expressed biosynthetic gene clusters while associated with particles.

Despite the generally small size of free-living marine prokaryote genomes, diverse sets of BGCs have been previously reported from free-living marine prokaryotes[39]. The production and potential release of secondary metabolites by free-living prokaryotes has received little consideration so far. This may be primarily due to the perception that bioactive secondary metabolites would be ineffective in dilute planktonic environments, and, thus, not confer the selective advantages experienced by microbes associated with highly structured microhabitats (e.g., particles, sediments, and soils). Yet, analysis of the FL metatranscriptomes in this study revealed the expression of BGC transcripts associated primarily with *Alphaproteobacteria*, Omnitrophota, SAR324, and Desulfobacterota

MAGs. The transcripts were predominantly from terpene, non-ribosomal peptide, and lactone clusters with inferred antibiotic activity, as well as roles in oxidative stress and cell-to-cell signaling. Notably, MAGs with an apparent FL preference ($P < 0.05$; FDR = 5%) expressed BGC genes in both the FL and PA fractions with a high degree of overlap (Supplementary Fig. 5a–c). We postulate some of the free-living cells, and thus their transcripts, could have been captured by the 2.7 µm PA filters during sampling. It is also possible some MAGs more abundant in the FL fraction dissociated from particles during sample processing. Microorganisms likely also attach to, and disassociate from, particles as they sink through the water column. Some cells that associate with particles in the surface ocean may remain trapped within particles as they sink into realms that no longer favor their survival in the FL state. These hypotheses require further investigation, as do other possible roles these secondary metabolites might play in the free-living state, including grazer avoidance. Bacterial MAGs with high levels of BGC expression in both sample types were affiliated with Desulfobacterota, Omnitrophota, and Planctomycetota. The presence of MAGs from these phyla in both FL and PA, and the robust expression of BGC transcripts in both fractions may indicate these taxa interact intermittently with particles.

Archaea have only recently gained attention for their potential to produce secondary metabolites[30,40,41]. Fifty-eight BGCs affiliated with Nanoarchaeota, Thermoplasmatota, Aenigmarchaeota, Altiarchaeota, Halobacterota, Crenarchaeota, and Undinarchaeota (previously UAP2) MAGs were detected and encoded primarily terpene and polyketide synthases, as well as RiPPs and NRPSs (Fig. 1b). Thirty-one clusters were identified within Nanoarchaota MAGs, a group reported previously to possess biosynthetic genes for molecules with putative antibiotic

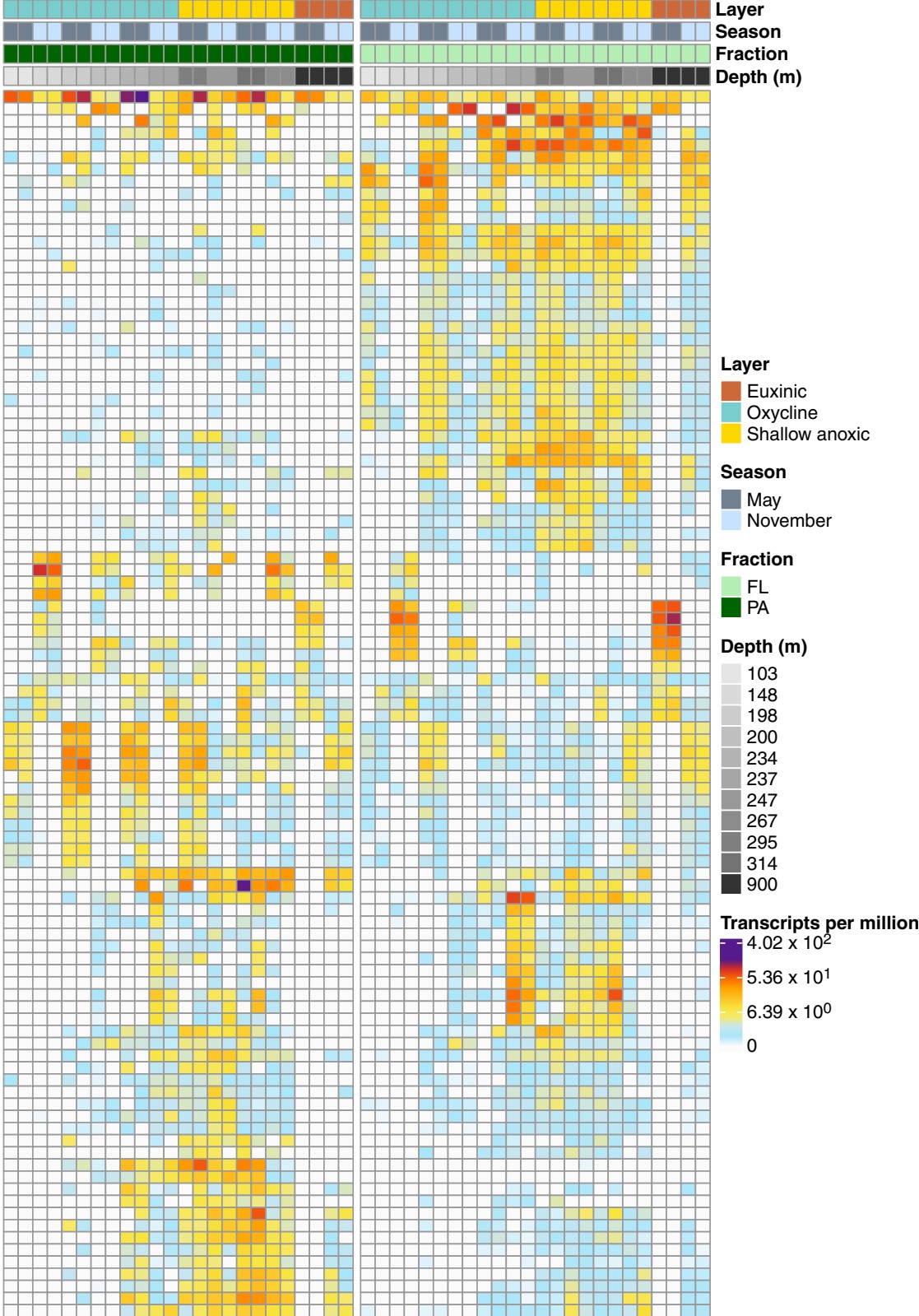

**Fig. 2 | Expression of secondary metabolite biosynthetic transcripts from individual MAGs in metatranscriptomic samples.** Each row represents a biosynthetic transcript, each column represents a sample from the PA fraction (left) and FL fraction (right), and the color represents the log-normalized transcripts per million. Source data are provided as a Source Data file.

properties[42]. Metatranscriptomics revealed NRPS and terpene cluster expression from FL oxycline samples from six FL-abundant Nanoarchaeota MAGs. Terpene and polyketide synthase transcripts from four Thermoplasmatota MAGs with no fraction preference were also expressed at oxycline, shallow anoxic, and euxinic depths in PA and FL samples. Further studies of these clades may provide a better ecological understanding of these archaea and new natural product discoveries.

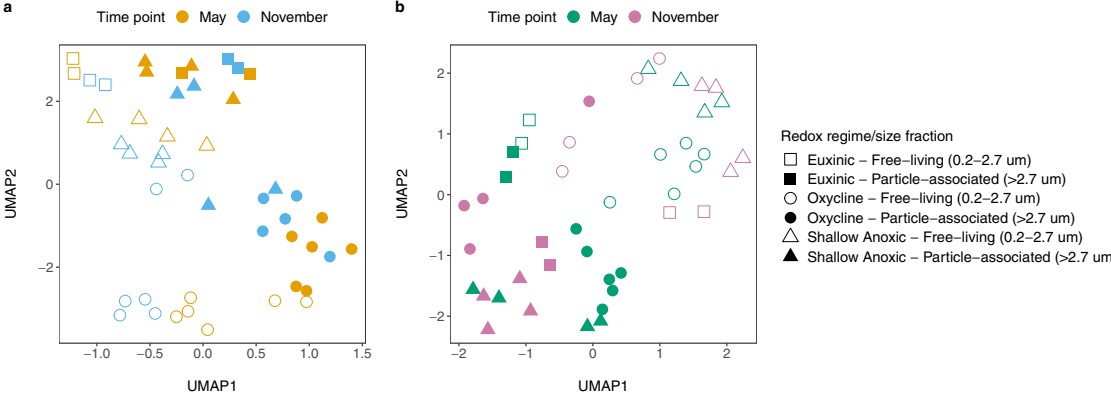

**Fig. 3 | Uniform manifold approximation and projection (UMAP) analysis of metagenomic and metatranscriptomic reads recruited to BGCs.** UMAP analysis for read mapping data of particle-associated and free-living metagenomes (**a**) and metatranscriptomes (**b**) to all BGCs longer than 10 kb total length detected in Cariaco MAGs. Each point represents the BGC expression profile in a sample, with redox regimes denoted by different shapes. The two size fractions are represented by filled-in and hollow shapes, and sampling time points are colored differently. Source data are provided as a Source Data file.

## Distribution and expression of BGCs across gradients

We observed differences in BGC abundance and expression across size fractions in both the metagenomic and metatranscriptomic samples (Fig. 3a, b). Uniform manifold approximation and projection (UMAP)[43] analysis of metagenomic and metatranscriptomic read recruitment to biosynthetic clusters primarily separated PA from FL sample types in most datasets (Fig. 3a, b). For the same size fraction and redox regime, UMAP analysis further separated most datasets between the two sampling points (May vs. November), particularly for BGC expression in oxycline and euxinic water features.

## Ladderane biosynthetic cluster detection

We detected ladderane BGCs in bacterial MAGs affiliated with Desulfobacterota, Fibrobacterota, Myxococcota, Verrucomicrobiota, Planctomycetota, Latescibacterota, Omnitrophota, GCA-001730085, and UBP3. Ladderane lipids are strictly associated with bacterial genera within the Planctomycetota phylum that perform anammox, but the pathway of ladderane biosynthesis and associated enzymes is unknown. BGCs that resemble ladderane clusters have been reported for non-Planctomycetota genomes[44], but an association between those and the presence of ladderane lipids was not made. Assessment of contigs containing ladderane BGCs by GUNC[45] could not identify any contaminated or chimeric contigs. Clusters annotated as ladderanes were expressed by all the phyla to which they were attributed (Supplementary Fig. 6). The two Planctomycetota MAGs that expressed ladderane clusters were differentially abundant in the FL fraction and were from the anammoxer genus *Scalindua*. The highest expression was observed at shallow anoxic depths (Supplementary Fig. 6). We conclude that the *Scalindua* ladderane clusters were accurately annotated, based on prior knowledge of anammoxers lipids and our expression profiles. Clusters of remaining MAGs encoding ladderanes may serve unknown functions in Cariaco Basin. Plausible in silico explanations for ladderanes in non-anammox taxa include possible involvement in fatty acid biosynthesis[44] and in lineage divergence of closely-related taxa via the acquisition of ladderane genes[46]. These could apply to the Cariaco Basin but needs to be validated experimentally.

## Oxidative stress genes in biosynthetic gene clusters

We annotated genes within 118 BGCs (primarily RiPPs, terpenes, NRPSs, and lactones) encoding for proteins that detoxify, promote biofilm formation[47], or counter damage from free radicals. These BGCs were primarily associated with *Alphaproteobacteria*, Desulfobacterota, Omnitrophota, Planctomycetota, and Myxococcota MAGs. Oxidative stress-related genes from these clusters were functionally annotated mostly as alkyl hydroperoxide reductase subunit C (*ahpC*), glutathione S-transferase (*gst*), and nickel superoxide dismutase. It is possible these enzymes to assist in the intracellular regulation of the free radicals' concentrations, albeit previous studies found AhpC and GST to contribute directly to secondary metabolite biosynthesis[48,49]. In FL metatranscriptomes, transcripts associated with oxidative stress from lactone, phosphonate, and terpene clusters were primarily expressed by FL-abundant *Scalindua* Planctomycetota, Chloroflexota, and SAR324 MAGs from oxycline and shallow anoxic depths, habitats that exhibit oxygen fluctuations.

To further identify redox-related compounds in the Cariaco BGCs, we compared them to the MIBiG[50] database, which contains community-curated clusters with known functions. Cellular-level redox-cycling antibiotics can infiltrate and impose oxidative stress on target cells[45]. As an example, four of the BGCs contained genes encoding phenazine or phenazine-like biosynthesis proteins. Phenazines are redox-active compounds known to contribute to the formation of bacterial biofilms and to cause debilitating oxidative stress in targeted cells by forming intracellular free radicals of both reactive oxygen and nitrogen species[51,52]. Nevertheless, the expression of phenazines could increase microbial fitness in Cariaco Basin by enhancing phosphorus cycling. Within the redoxcline of the Cariaco Basin exists a challenging variability in phosphate concentrations whose fate (precipitation vs. remobilization) is controlled by the delivery of iron and manganese in the water column[53]. Phenazines are phosphorus/iron-regulated antibiotics suggested to promote microbial growth under phosphorus starvation via solubilization of phosphates through the reduction of iron oxides[54]. Expression of genes associated with redox-cycling antibiotics was found primarily in FL metatranscriptomes at all water layers.

Genomes encoding clusters with antibiotic properties often contain genes coding for proteins within the same cluster that prevents self-toxicity[55]. We applied Antibiotic Resistant Target Seeker (ARTS[33,34]) to detect antibiotic resistance genes in our BGCs. We detected only 4 types of proteins/protein domains involved in resistance (Supplementary Data 5). These include 13 MAGs that had ABC and RND efflux pumps, RND-type membrane proteins of the efflux complex MexW/MexI/MexH[56], and 8 MAGs that contained pentapeptide repeats[57]. These were all associated with BGCs that coded for terpenes, bacteriocins, T1PKS/T3PKS, homoserine lactone (hserlactone), NRPS/NRPS-like, beta lactones, aryl polyenes, and hglE-KS.

## Core biosynthetic genes and tailoring enzymes in Cariaco biosynthetic gene clusters

Analysis of antiSMASH results revealed the presence of core biosynthetic genes that are highly conserved and essential to secondary

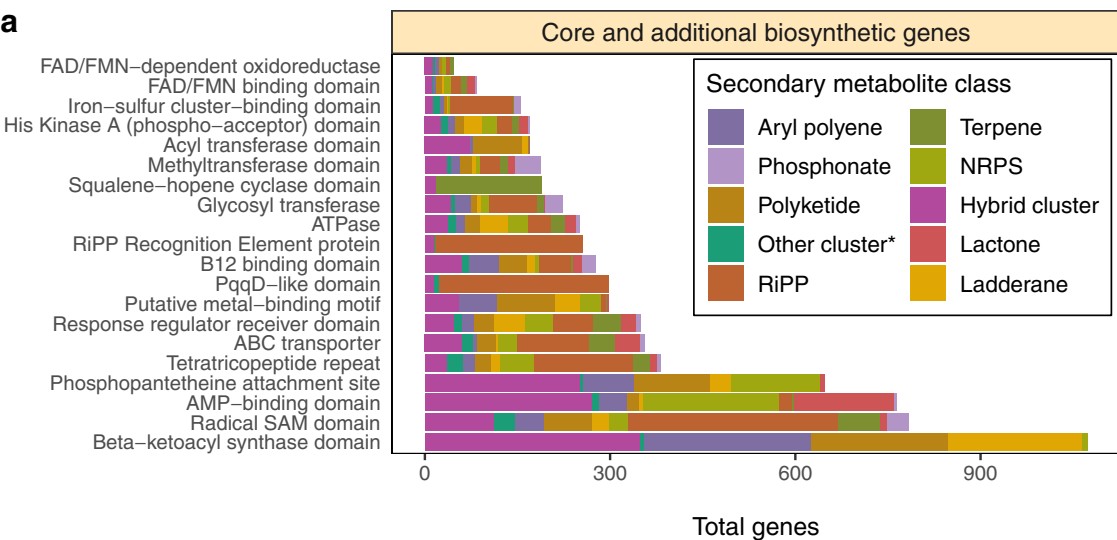

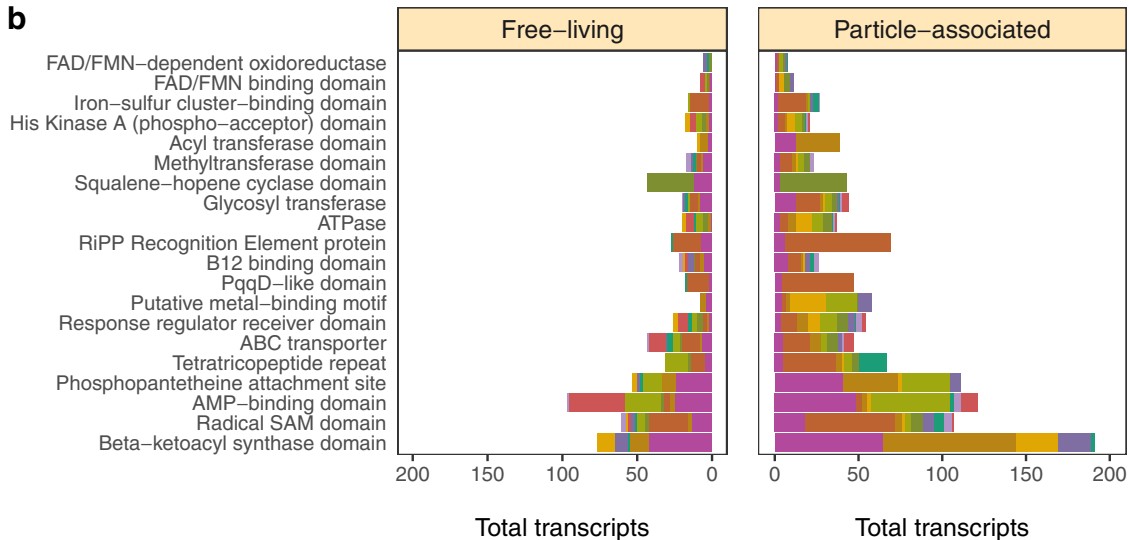

**Fig. 4 | Distribution of core and additional biosynthetic genes or domains and transcripts from biosynthetic gene clusters. a** Distribution of the most frequently detected core/additional biosynthetic genes, genes encoding tailoring enzymes, and biosynthetically important protein domains in clusters ≥10 kb in length. The "Core and additional biosynthetic genes" strip title of **a** refers to genes or domains. **b** Core/additional biosynthetic transcripts, as well as transcripts encoding tailoring enzymes and biosynthetically important protein domains differentially expressed or solely expressed in the free-living (left-hand panel) and particle-associated (right-hand panel) metatranscriptomes; colored by BGC product class. Source data are provided as a Source Data file.

metabolite biosynthesis. The most frequently detected core biosynthetic gene encoded β-ketoacyl synthase, an essential enzyme in fatty acid biosynthesis[58] (Fig. 4a). We also detected pyrroloquinoline quinone subunit D-like (PqqD-like) synthase, which is essential to the biosynthesis of RiPP-recognition element-dependent RiPPs[59] (Fig. 4a). In the PA fraction, there were 235 differentially expressed or uniquely expressed PqqD-like and beta-ketoacyl synthase transcripts (Fig. 4b), while only 94 were detected in the FL fraction (Fig. 4b).

Various tailoring enzymes identified in the recovered Cariaco BGCs suggest the implementation of diverse chemical transformations and post-translational modifications. Oxygen availability along the oxycline in Cariaco Basin could impact the distribution/number of BGCs and tailoring enzymes utilizing molecular oxygen. We searched the BGCs for tailoring enzymes, including Rieske non-heme iron oxygenases (ROs). These enzymes contain oxygen-sensitive [2Fe-2S] clusters and are involved in the synthesis of bioactive natural products[60]. Overall, we detected eight types of ROs in 32 MAGs encoding BGCs for terpenes, beta lactones, T1PKS/T3PKS, phosphonates, RiPPs (lasso/thio/ranthipeptides, linear azole/azoline-

containing peptides), NRPS (cyclodipeptides) and RiPP- and NRPS-like clusters. These ROs/ROs-domains were annotated to dioxygenases associated with degradation of aromatic amino acids (tyrosine/tryptophan), phosphonate and sulfur (taurine) cycling, pigment biosynthesis (carotenoids/betalain), and glyoxalase/bleomycin/validamycin dioxygenase superfamilies. This suggests that the identified ROs/ROs-domains can be directly (e.g., synthesis of pigments and antibiotics) or indirectly (via nutrient/amino acid cycling) involved in the synthesis of these secondary metabolites.

Other tailoring enzymes in BGCs included radical SAM proteins, glycosyl transferases, and flavoenzymes (Fig. 4a). The most abundant of these were radical SAM proteins, which are known for imparting diverse post-translational modifications on RiPPs[61]. Post-translational glycosylation of secondary metabolites by glycosyl transferases can have a variety of effects, such as toxicity reduction for the producer of the metabolite[62]. Flavoenzymes help tailoring structurally diverse secondary metabolites through various redox reactions, including single-electron transfers[63]. A total of 169 transcripts encoding the above tailoring enzymes were differentially expressed or uniquely

expressed in the PA fraction, compared to 94 transcripts in the FL samples (Fig. 4b).

Genes encoding specialized domains involved in peptide biosynthesis were also detected in the Cariaco biosynthetic clusters (Fig. 4a). A prevalence of $B_{12}$-binding domains identified in biosynthetic genes raises the possibility of $B_{12}$-dependent methylation during synthesis or post-translational modification of biosynthesized peptides[61,64]. Phosphopantetheine attachment site domains were also ubiquitous in Cariaco BGCs. Phosphopantetheine prosthetic groups from acyl carrier proteins are transferred by acyl transferases[65], both of which were numerous in Cariaco clusters. Tetratricopeptide repeats were prevalent as well, which can mediate protein–protein interactions in diverse cell processes[66] by binding many distinct types of ligands[67]. The presence of various biosynthetically important protein domains present in the recovered BGCs suggests a variety of diverse chemical transformations and post-translational modifications that could shape the secondary metabolites synthesized by the particle-associated and free-living microbes identified in the water column of Cariaco Basin.

## Discussion

We performed genome mining to detect and classify BGCs across a diverse set of bacterial and archaea phyla recovered from the anoxic depths of the Cariaco Basin. Although the increasing number of available genomes and bioinformatic approaches have revolutionized the discovery of secondary metabolites[5], a key issue that remains is linking the detected clusters to biological activity. Previous studies of marine Planctomycetota showed that aqueous and organic extracts of isolates that contain bioinformatically-predicted BGCs exhibited antimicrobial and antifungal activity[37]. However, the vast majority of microorganisms, particularly those from challenging environments like oxygen-depleted systems, escape cultivation, thus hindering our ability to use similar approaches to explore the bioactive potential of predicted BGCs. Recently, an analysis of >1000 publicly available marine metagenomes revealed ~40,000 putative BGCs[4]. Nonetheless, this analysis did not include samples from sulfidic waters. For this environmental survey of BGCs, we used mapped metatranscriptomes collected and preserved in situ to our BGCs to unveil the expression profiles of detected biosynthetic clusters, and to investigate the potential role of redox conditions and particles in the observed patterns.

Particles provide colonizable, nutrient-rich substrates where metabolites can be concentrated and exchanged and can provide protection for oxygen- or sulfide-sensitive microbiota. Previous work shows particle-associated microbial assemblages from the Eastern Tropical North Pacific oxygen minimum zone possess genes coding for antibiotic resistance, motility, cell-to-cell transfer, and signal recognition[67,68], and microorganisms are able to proliferate in particles in suboxic to anoxic zones where reducing conditions can persist for extended periods of time[69,70]. Consistent with previous culture-based studies, BGCs may allow PA taxa to compete for precious resources, prevent the growth of other potential particle colonizers, and aid survival in oxygen-depleted conditions[36]. Our study revealed enhanced expression of BGCs by members of Myxococcota, Desulfobacterota, Omnitrophota, Planctomycetota, and *Gammaproteobacteria* within the PA fractions.

Analysis with UMAP of biosynthetic cluster abundances and expression profiles revealed a marked separation between the PA and FL size fractions in both the metagenomic and metatranscriptomic data. The niche preferences of taxa behind the MAGs we recovered, as well as the two different sampling times, likely play a role in the observed differences in expression profiles of biosynthetic clusters in our PA vs. FL samples. We detected differences between the sampling season and redox regime within the metagenomes. In the metagenomic samples, the PA euxinic and deep anoxic samples, as well as the

FL euxinic samples, clustered together (Fig. 3a). The abundance of metagenome reads mapped to MAGs across size fractions was similar at depths where oxygen is very limited or absent, and contributed to the clustering of BGC read abundances within these samples. The FL shallow anoxic and oxycline, and the PA oxycline samples formed three distinct clusters, suggesting differing redox conditions shaped BGC composition and abundance in these samples. Within oxycline samples, the influence of oxygen and the separation between PA and FL size fractions is evident.

UMAP analysis of the metatranscriptomic data revealed BGC expression profiles that differentiated primarily by size fraction as well as the season of sampling (Fig. 3b). Some overlap is observed between BGCs expressed in the PA and FL fractions, consistent with the idea that some taxa may transiently associate with particles as they sink through the water column. Paoli et al. (2022)[4] examined MAGs recovered from PA and FL fractions in global datasets (that did not include sulfidic end-members) and they found genes for terpenes and RiPPs enriched in the FL fraction, and NRPS and PKS genes enriched in PA samples. This supports the idea that taxa and the genes they carry are shaped by their FL vs. PA lifestyle (niche requirements). Seasonal differences in primary productivity can also shape microbial communities and the genes they express. In Cariaco Basin, upwelling of nutrients occurs between January and March, fueling increased primary productivity[71]. This may be a contributing factor to the observed separation of most PA vs. FL BGC profiles (Fig. 3a, b) because in the FL state, microorganisms will experience environmental shifts more directly than those protected within particles.

While little is known about secondary metabolite expression in free-living marine prokaryotes, the biosynthetic potential is known to be widespread in their genomes[39]. We detected expression of BGCs in the FL metatranscriptomes, predominantly from *Alphaproteobacteria*, Omnitrophota, SAR324, and Desulfobacterota MAGs. The overlap in BGC expression detected in the PA and FL transcriptomes mapping onto preferentially FL-associated MAGs was unique, as we did not observe the same phenomenon in the expression pattern of preferentially PA-associated MAGs. This supports our hypothesis that the PA metatranscriptomes captured some of the BGC expression signals from the FL samples. While it is less clear how free-living microbes could benefit from the release of secondary metabolites, we conclude that interactions with particles alone cannot account for all the expression of biosynthetic transcripts in the FL samples. Some of these compounds (such as the ladderanes), likely only serve intracellular roles within free-living prokaryotes. It is also plausible that higher expression of BGCs in PA samples reflects a more commonplace release of secondary metabolites within particles than within free-living ODWC microbial populations. The role of secondary metabolites in microbial fitness is an open debate because possession of secondary metabolism can enhance overall fitness, but not all products of secondary metabolism will necessarily have an effect on the producer[72]. Nonetheless, secondary metabolites are reported to affect niche utilization, shape microbial community assembly, and act as a functional trait driving ecological diversification among closely-related bacteria inhabiting the same microenvironments[73,74]. Likewise, the example of phenazines and phosphorus acquisition can be a paradigm of dual/pleiotropic functions of secondary metabolites where they can serve as potential antibiotics and regulators of nutrient cycling. Colocalization of antibiotic resistance genes within biosynthetic clusters has been previously observed[75]. Bacteria may have evolved pleiotropic switching capabilities that allow simultaneous expression of secondary metabolites with other co-localized genes in a cluster (e.g., antibiotic regulation and resistance) as a survival strategy under unfavorable conditions, and as a self-protection mechanism[55]. Co-localized genes encoding antibiotic resistance were present in the BGCs we identified. In addition to antibiotic resistance genes, clusters contained a diverse array of core biosynthetic synthases, tailoring

enzymes, and significant protein domains involved in secondary metabolite synthesis and post-translational modifications.

In summary, our investigation of BGCs in metagenomic and metatranscriptomic datasets from an oxygen-depleted marine water column provides considerable evidence for secondary metabolite synthesis over a wide taxonomic distribution from 44 bacterial and 8 archaeal phyla. More BGCs were expressed (particularly coding for non-ribosomal peptides, polyketides, RiPPs, and terpenes) by taxa whose MAGs were particle-associated than in the free-living fraction. The BGCs identified here hint at a complex network of ecological interactions coupled with a competitive, yet communicative lifestyle mediated by chemical and toxin production not only within PA microbes, but to a lesser extent, within the FL communities. These findings open the door for future laboratory characterization of genes for novel bioactive metabolites with potential ecological and pharmaceutical importance.

## Methods

### Sample collection

Water samples for metagenomic analyses were collected from 6 depths during two cruises in May 2014 and in November 2014 using Niskin bottles, as described in detail in ref. [21] (Supplementary Data 6). Specifically, 8–10 L water samples for metagenomic analysis were gravity-filtered sequentially through EMD Millipore 2.7 μm glass fiber membranes 47 mm diameter (PA fraction), and then through 0.2 μm Sterivex filters (FL fraction) and stored frozen at −20 °C in the field and then −80 °C in the laboratory until extraction. Water samples were also collected and preserved in situ for isolation of RNA and construction of metatranscriptome libraries from depths selected to capture anoxic and sulfidic water layers. RNA sample collections were conducted with a "Microbial Sampler−Submersible Incubation Device" (MS-SID)[76,77]. Water (2 L) was sequentially filtered through EMD Millipore 2.7 μm glass fiber filters and then through 0.2 μm Millipore Express polysulfone membranes at depth. The filters were preserved immediately in situ with RNAlater®. Upon MS-SID retrieval, preserved filters were transferred to cryovials with additional RNAlater and stored frozen at −20 °C in the field and then −80 °C in the laboratory until extraction. Biogeochemical data collected in support of molecular samples are available at https://www.bco-dmo.org/dataset/652313/data.

### DNA extractions and sequencing

DNA was extracted from all samples according to ref. [78] and ref. [68] and described in detail in ref. [22]. Briefly, lysozyme solution (2 mg in 40 μL of lysis buffer) was added directly to the tube containing the 2.7 μm membrane filter or to the Sterivex cartridge, and was incubated for 45 min at 37 °C. Subsequently, Proteinase K solution (1 mg in 100 μl lysis buffer, with 100 μl 20% SDS) was added, and then incubated for 2 h at 55 °C. The lysate was transferred to a clean tube and nucleic acids were extracted once with phenol:chloroform:isoamyl alcohol (25:24:1) and once with chloroform:isoamyl alcohol (24:1). The aqueous phase was concentrated using Amicon Ultra-4w/100 kDa MWCO centrifugal filters (Millipore). After extraction, DNA was purified with the Genomic DNA Clean and Concentrator-25 kit (Zymo Research), eluted into 10 mM Tris-HCl, and frozen until downstream analysis. Aliquots of the extracted DNA were sent to the Georgia Genomics for library preparation and paired-end 2 × 150 bp Illumina NextSeq sequencing. The R1 and R2 reads were filtered using Trimmomatic[79] 0.39. Trimmomatic performs a "sliding window" trimming, removing sequence data when the average quality within the window (eight nucleotides used here) drops below a threshold (set to 12). The length of the trimmed sequences was set to a minimum of 50 nucleotides.

### RNA extraction and sequencing

RNA was extracted using a modification of the mirVana miRNA Isolation kit (Ambion, Life Technologies, Carlsbad, CA, USA) as in ref. [80].

Briefly, filters were thawed on ice and the RNA stabilizing buffer (RNAlater) was removed from the cryovials. Cells on filters were lysed by adding lysis buffer and miRNA homogenate additive (Ambion) into the cryovial or cartridge. After vortexing and incubation on ice, lysates were transferred to RNAase-free tubes and processed using an acid−phenol/chloroform extraction following the manufacturer's suggestions. We used a TURBO DNA-free kit (Ambion, Foster City, CA, USA) to remove carryover DNA and we purified the extracts using the RNeasy MinElute Cleanup Kit (Qiagen, Hilden, Germany). Removal of DNA was confirmed by PCR using the forward primer (5′-AYTGG-GYDTAAAGNG-3′) and a mix of reverse primers (5′-GCCTT GCCAGCCCGCTCAG, TACCRGGGTHTCTAATCC, TACCAGAGTATCT AATTC, CTACDSRGGTMTCTAATC, and TACNVGGGTATCTAATCC-3′ in a 6:1:2:12 ratio, respectively), designed to cover most hypervariable regions of bacterial 16 S rRNA (ref. [81]). cDNA libraries were prepared using the ScriptSeq RNA-Seq Library Preparation Kit (Illumina). Excess nucleotides and PCR primers were removed from the library using the Agencourt AMPure™ XP (Beckman-Coulter) kit. The Illumina NextSeq platform was used for paired-end 2 × 150 sequencing at the Georgia Genomic Facility. The R1 and R2 reads were filtered using Trimmomatic.

### MAG co-assembly, binning, and taxonomic assignment

Metagenomes originating from adjacent regions (such as geographic regions or, in this case, adjacent depths targeted in this study) are likely to overlap in the sequence space, increasing the mean coverage and extent of reconstruction of MAGs when using a co-assembly approach. In order to reconstruct MAGs, the trimmed reads of metagenomic datasets from the anoxic to sulfidic depths (314 and 900 m in May, 267 and 900 m in November) and both PA and FL fractions were co-assembled into contigs using SPAdes 3.11.1[82] with default values and flag "−meta". Assembled contigs were binned using MetaBAT 2.12.1[83] with default values. CheckM 1.0.1161[84] was used to estimate the completeness and contamination of the reconstructed genomes. Only MAGs with ≥75% complete and ≤5% contamination were used for the downstream analysis (Supplementary Data 1). The taxonomic placement of the MAGs was performed with GTDB-Tk[85] 2.1.1. The taxonomic identification of the recovered MAGs revealed the presence of three known lab contaminants, including Burkholderia contaminans[86] that was reconstructed in the first marine genomic study.

### Removal of redundant MAGs

To collapse any redundancy, a workflow using Anvi'o 4[87] was implemented as described in ref. [88]. Scaffolds from all MAGs were concatenated into a single FASTA file for mapping and processing. Anvi'o commands "anvi-gen-contigs-database" and "anvi-run-hmms" were run with default parameters to create a MAG contig database, and scan MAGs with HMMs, respectively. Contigs were then used to recruit short reads from all metagenomic samples using the Bowtie2[89] 2.5.0 commands "bowtie2-build" and "bowtie2" with default parameters, and SAMtools[90] 1.16.1 commands "view", "sort", and "index" to convert resulting Sequence Alignment/Map format (SAM) files into Binary Alignment/Map (BAM) format, as well as sort and index the converted BAM files. The Anvi'o command "anvi-merge" was implemented to create a merged profile database of all MAGs, describing the distribution and detection statistics of scaffolds in MAGs across all the metagenomic samples. The average nucleotide identity (ANI) and Pearson correlation values were then calculated to identify MAGs with high sequence similarity and those that were distributed similarly across metagenomes. Pairwise Pearson correlations of the MAGs were calculated using the "cor" function in R[91], and ANI values were calculated for MAGs using NUCmer (from the MUMmer[92] 3.23 package) with default settings, grouped by their taxonomy, such that ANI was not computed for pairs of MAGs which did not belong to the same phylum. Anvi'o scripts were then used to classify MAGs as redundant if (1) their

                               

ANI was at least a 98% match (with a minimum alignment of 75% of the smaller genome in each comparison) and (2) the Pearson coefficient for their distribution across datasets was >0.9.

## Calculation of MAG relative abundances

Reads from 48 metagenomic samples (two sample types: PA- and FL-fraction, two replicates per fraction from six depths, taken over two sampling time points = 48 metagenomes; Supplementary Data 6) were mapped to each MAG using the BWA[93] 2.0 aligner via the CoverM[94] 0.6.1 command line tool. The CoverM tool automatically concatenated all the MAGs into a single file, and metagenome reads were recruited to MAG contigs, setting the parameter –min-read-percent-identity to 95 and –min-read-aligned-percent to 50. The "Relative Abundance" CoverM method on the "genome" setting was used to calculate the relative abundances of the 565 MAGs in each of the metagenomic samples. Each relative abundance was calculated as the percentage of total reads from the sample that uniquely mapped to a MAG (with consideration for the percentage of unmapped reads in the sample). The relative abundance values were log-transformed using the log1p formula: $y = \log(x + 1)$ and were used for heatmap plotting.

## Assessing BGC-containing contigs for chimerism and contamination using GUNC

The GUNC[45] command line tool was used with default settings to assess contigs from Cariaco MAGs for contamination that contained ladderane BGCs predicted by antiSMASH[19]. The flat file outputs from running GUNC 1.0.5 on all contigs containing BGCs were then manually assessed for potential chimerism and/or contamination.

## Differential abundance analysis

Differential abundance (PA- or FL-abundant) was determined for individual MAGs exhibiting differences in abundance between sample type (PA, FL) and water layer (oxycline, shallow anoxic, euxinic). Reads were recruited to a concatenated FASTA file containing whole genome contigs using CoverM[94]. A counts matrix was created with rows containing individual MAG read mapping counts and with metagenomic samples as columns. Count data for all MAGs was analyzed to calculate DESeq2 1.34.0 size factors for cross-sample count normalization. The differential abundance of MAGs between fraction sizes and water layers was modeled using the DESeq2 negative binomial model with the metadata variables of fraction size and water layer in which "count" was the dependent variable and "fraction" as well as "water layer" were independent variables. The significant differential abundances of MAGs (with an FDR-corrected $P < 0.05$) identified by comparing the PA and FL samples were grouped by water layer, and direct comparisons were made between normalized counts of significantly differentially abundant MAGs from the oxycline, shallow anoxic, and euxinic depths.

## Similarity clustering of BGCs using BiG-SCAPE

The redundancy of the predicted biosynthetic cluster sequences recovered from the Cariaco MAGs was assessed using the BiG-SCAPE[32] 1.1.4 command line tool with default parameters. The resulting Gene Cluster Families (GCFs) from this sequence similarity network analysis were visually assessed using BiG-SCAPE's default index.html output file.

## Scanning of MAGs for BGCs, functionally annotating genes within clusters, and comparing mined clusters to the MiBIG database BGCs

Genes were predicted using Prodigal[95] 2.6.3 for all 565 MAGs. The resulting genes of each MAG were individually scanned for BGCs using antiSMASH 6[19] with default parameters. Gene clusters with a total length of less than 10 kb were discarded from downstream analysis to minimize the inclusion of fragmented BGCs in our data.

The genes predicted using Prodigal were scanned using the InterProScan 581[96] and Prokka[97] 1.14.6 command line tools with default parameters for functional annotations, as well as during the implementation of the antiSMASH 6 pipeline with the antiSMASH HMM databases. We manually searched the resulting annotations for genes and domains that encoded a variety of functions, such protein domains involved in post-translational modifications. The results of comparing the mined Cariaco BGCs to the MiBIG database BGCs was scraped using R from the output HTML files from scanning each MAG with antiSMASH.

## Detection of antibiotic resistance genes in BGCs using ARTS

The presence of putative antibiotic resistance genes was examined with ARTS version 2[33,34] that implements antiSMASH 5.0. The web interface was used. The results in the "Proximity: BGC table with localized hits" were manually inspected. The location and the annotation of potential antibiotic resistance genes that show colocalization with BGS is recorded in Supplementary Data 5.

## Metatranscriptomic read mapping of RNA-seq data to BGCs

Metatranscriptomic samples were individually mapped to the concatenated gene FASTA file using the minimap2[98] 2.24-r1122 sequence alignment algorithm with default parameters. The resulting output files in PAF format were manually filtered of supplementary alignments using a custom R script, and only alignments incorporating at least 50% of the length of a read pair with at least 95% percent identity were retained. The same custom script was utilized to concatenate all individual alignment counts into a single file in a matrix format, with each sample representing a column and each row representing RNA-Seq alignment counts to a gene. The metatranscriptomic counts' matrix was normalized to transcripts per million (TPM) and the values were log-transformed using the log1p formula: $y = \log(x + 1)$ and were used for heatmap plotting.

## Differential gene expression analysis

To detect differential expression of individual genes within differently expressed biosynthetic clusters between sampling depths, the read counts matrix was modeled in the context of the metadata variable size fraction using a negative binomial model implemented with DESeq2 1.34.0 in R. Count data for all genes from all MAGs was analyzed independently so that the DESeq2 size factors for cross-sample count normalization would reflect the total transcriptomic activity of MAGs in each sample. This approach is robust to biases in total transcriptomic activity per organism between samples, and is used to identify differences in gene expression independent of changes in taxonomic composition, similar to previously reported methods[18,99]. After size factor normalization, read counts were fit to a negative binomial model in which "count" was the dependent variable and "size fraction" was an independent variable. To test whether any genes exhibited differential expression associated with different size fractions, the differential expression results were saved and analyzed. The significant genes (with an FDR-corrected $P < 0.05$) were identified by comparing the PA and FL samples and direct comparisons were made between normalized counts of genes that differed significantly in expression profiles. This method confirmed the differential expression of individual genes within each differentially expressed biosynthetic cluster.

## UMAP analysis on BGC abundances in metatranscriptomic and metagenomic datasets

The normalized abundances of BGC read mapping data from both metagenomic and metatranscriptomic read recruitment using minimap2 were used as input for UMAP analysis in R using the umap[100] 0.2.9.0 package (https://github.com/tkonopka/umap). The results were plotted using ggplot2[101] 3.3.6. Clustering of the UMAP embedding

                                                                                        

was done using the hierarchical density-based spatial clustering (HDBSCAN) function from the dbscan[102] 1.1-11 package.

## Reporting summary
Further information on research design is available in the Nature Portfolio Reporting Summary linked to this article.

## Data availability
The metatranscriptome and metagenome data generated in this study have been deposited in the NCBI database under accession code PRJNA326482. The processed metagenome-assembled genomes (in FASTA format) and biosynthetic gene cluster files (in ZIP format) are available at OSF [https://osf.io/usm8r/]. The biogeochemistry data from the CARIACO Basin Time Series Station for May to November 2014 are available through the Biological and Chemical Oceanography Data Management Office (BCO-DMO) at the Woods Hole Oceanographic Institution [https://www.bco-dmo.org/dataset/652313/data]. The MIBiG 2.0 database is publicly available [https://mibig.secondarymetabolites.org]. Source data are provided with this paper.

## Code availability
The scripts used for all bioinformatic pipelines, data processing, and plotting used in this study are available in the following GitHub repository: https://github.com/d-mcgrath/cariaco_basin[103].

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

## Acknowledgements

We thank the staff of Fundación La Salle de Ciencias Naturales (FLASA), EDIMAR, Porlamar, Edo Nueva Esparta, Venezuela, and the crew of the R/V *Hermano Ginés* for their support during fieldwork for this study, especially Y. Astor and R. Varela. The fieldwork that provided samples and data for this study was supported by National Science Foundation (NSF) grants (OCE-1336082 to V.E. and OCE-1335436 and OCE-1259110 to G.T.T., Stony Brook University). Analysis of the data were partially supported by NSF grant OCE-19924492 to M.P. and V.E. and Simons Foundation award 929985 to M.P.

## Author contributions

M.P., V.E., and G.T.T. designed the original research project. M.P., V.E., D.G.-M., and P.M. designed the secondary metabolite study. G.T.T., M.P., E.S., and V.E. sampled the ecosystem. M.P. conducted DNA and RNA extractions. M.P. assembled the metagenomes and binned metagenome-assembled genomes. D.G.-M. performed genome-resolved metagenomics and metatranscriptomics. P.M. manually curated the meta-omics data, and P.M. and D.G.-M. conducted the analysis of secondary metabolite data. P.M. and D.G.-M. wrote the paper with input from M.P. and V.E. G.T. and E.S. contributed to the final manuscript.

## Competing interests

The authors declare no competing interests.
