## [Peer Review File · Nature Communications]

Diverse secondary metabolites are expressed in particle-associated and free-living microorganisms of the permanently anoxic Cariaco BasinReviewer #1 (Remarks to the Author):

McGrath et al. dives into the understudied realm of the biosynthetic potential of the Cariaco Basin including a focus on particle-associated vs. free-living lifestyles of microbes across euxinic and anoxic gradients. Their methods for analysis follow the trend set by several publications (Crits-Christoph et al. 2018, Nature; Nayfach et al. 2020 Nature Biotechnology; Paoli et al. 2022 Nature) for computational detection of biosynthetic gene clusters (BGCs) in MAGs here focusing on a somewhat unique habitat and microbial community composition relative to these previous studies. The overlay of metatranscriptomic data lends ecological relevance and we commend the authors on the inclusion of expression data in their study. However, the limited depth of their analysis and current unavailability of the code in a public repository (GitHub repo was empty as of 26.07.22) raises several concerns. See detailed comments below:

Note to authors: continuous line numbers including the main text have been added so as to make the location of suggested edits more specific. A PDF with these line numbers is uploaded for reference. Please include line numbers in further versions of this manuscript.

- The use of the word "high-quality" for the MAGs is up for debate here. Please correct me if standards have changed however according to the minimum information about a metagenome-assembled genomes (MIMAG) Bowers et al. Nat. Biotechnol. 35, 725–731 (2017) I believe high quality should be greater than or equal to 90% mean completeness and less than or equal to 5% mean contamination. See methods of Paoli et al. 2022 Nature for further clarity on how to stratify MAGs by quality.

- One major item of concern: the Github repository https://github.com/d-mcgrath/cariaco_basin/ is empty. This is unacceptable, given the code availability statement in the manuscript lists all code are available. Please push well-commented code with a detailed README.

- Line 79: "antimicrobial compounds (e.g., antibiotics, polyketides, RiPPs) – this sentence is somewhat misleading since many RiPPs and polyketides ARE antibiotics...perhaps NRPS would be more appropriate here?"

- Throughout the manuscript: there are several locations e.g., lines 23-24, "remains virtually unexplored", lines 68-69, lines 290-291 e.g., "for the first time in environmental surveys of BGCs, mapping of metatranscriptomes collected and preserved in situ to unveil..." where the authors claim to be "the first." Given the recent plethora of publications analyzing metagenomes/MAGS (Nishimura et al. 2022 Scientific Data) and in some cases from open ocean environments along oxygen/depth gradients and particle-associated vs. free-living (e.g., Paoli et al. 2022 Nature), we recommend the authors please tone down their language. While these statements of novelty may have been true at the outset of the study it is certainly no longer the 'first' to conduct this type of analysis.

- The authors highlight the oxycline as a key novel feature of their study. However the analysis devoted to how BGC composition changes along the oxygen gradient is underdeveloped. For example, does oxygen availability have any impact on the number of BGC-associated or tailoring enzymes utilizing molecular oxygen (e.g., Rieske oxygenases) encoded within BGCs? Similarly, under euxinic conditions do protein-coding genes in BGCs exhibit altered ratios of sulfur-containing amino acids?

- The discussion is remarkably short given the breadth of analysis that was carried out. Please consider amending with further discussion e.g., along the oxygen/sulfide gradients and other unique biogeochemical/taxonomic features of your sampling site.

- Lines 116-117: How redundant were the BGCs detected? How fragmented? In general the manuscript could use significantly more depth of analysis beyond a description, "which BGCs are there and expressed?" Rather than just reporting the total number of BGCs, comparative BiG-SLICE or BiG-SCAPE analysis would aid in the identification of similar and dissimilar biosynthetic gene cluster families across taxa and along the environmental gradients. Given that you have expression data, does analysis by BiG-MAP (Pascal Andreu et al., 2021) recover the same

differentially expressed BGCs you identified using DESeq2?

- Furthermore, you could do dimensionality reduction (e.g., UMAP) using metagenomic GCF abundances and overlay sample metadata to learn more about biosynthetic structure along your gradients e.g., see the methods section of Paoli et al. 2022 Nature for an example of such UMAP analysis.

- Lines 200-201, "Clusters of remaining MAGs classified by antiSMASH to encode ladderanes may synthesize other products, including polyunsaturated hydrocarbons." Can you quantify how many of the ladderane clusters have genes characteristic of PUFA clusters? PUFAs will have very distinct marker genes e.g., pfaA. I also believe they are more frequently classified as polyketide synthases than as ladderanes. Please verify the accuracy of this statement and expand.

- Line 220-221, the discussion of redox-active metabolites is lacking since redox-active do much more than oxidative stress..for example, iron or phosphorus cycling. Please discuss, in the context of McRose et al. 2021 Science and related work

- Lines 228-229 - Is there a reason you chose this more 'custom approach' for detection of antibiotic resistance genes in BGCs compared to using e.g., ARTS (Mungan et al. 2020 Nucleic Acids Research). How do your results compare to accepted tools for this analysis?

- Lines 309: "While it is less clear how free-living microbes could benefit from release of secondary metabolites..." - please discuss further here- some possible explanations and examples in the literature e.g., nutrient acquisition

- Finally, although I recognize this is not trivial and possibly not feasible given expense and time constraints, some form of experimental characterization of one or more BGCs or producing organism would be a valuable addition to validate computational predictions.

Reviewer #2 (Remarks to the Author):

This is an interesting study on BGC prevalence in permanently anoxic waters, and one of few studies to go beyond BGC potential and look at expression. Some comments below would be good for the authors to clarify to help improve the understanding of their study.

1. Page 3 line 70 – It was not completely clear to me why the roles of particles in the production of BGCs was a critical gap in knowledge. It may well be but thinks this needs to be better justified and explained in the text early on as to why it so important to study particles in relation to BGCs. How are particles important in this anoxic, sulfidic zone? Are they offering protection, a niche where metabolites can be concentrated and exchanged?

2. Page 3 line 71 – Perhaps instead of 'a large number' which is very subjective, use '565' or 'over 500' to add some level of detail here.

3. Page 3 line 76 – PA and FL need to be defined here and as per above the significance of analysing particles made clearer.

4. Page 3 line 83 – As per the established definition of MAG quality by the MIMAG (Minimum information about a metagenome-assembled genome) from Bowers et al (2017) I was under the impression that high quality MAGs needed to be closer to 90% bin completion. Could the authors clarify?

5. Page 4 line 102 – What is the definition of 'enriched' in this context? Enriched from what? Are they at higher numbers, abundances compared to others and if so what is the criteria and what values can be put to this if any?

6. Page 4 lines 116-118- In addition to terpenes, was there any evidence for siderophores, ectoines, or homoserine lactones production/capacity?

7. Page 5 line 132 – Re intercellular communication, as per the query above were any AHL-based

(or other) signalling systems observed?

8. Page 6 lines 163-164 – In terms of the overlap of BGCs between PA and FL fractions would it not be expected that at least some taxa would occupy both niches? I am not an expert in particle formation but would not free living marine microbes be the source of those residing on/in particles?

9. Page 7 line 198 - Can the authors postulate on any potential relevance to annamox at shallow oxycline depths?

10. Page 7 lines 205-215 – Do microbial communities associated with particles form biofilm-like phenotypes? If so there would be expected increased oxidative stress within the biofilm thus was there an increased prevalence of BGCs related to oxidative stress in particles (compared to free living)?

11. Page 22 line 443- I noted in the sampling that different samples were taken at different seasons. There was no real analysis and discussion on this and I am curious as to whether BGCs could be differentially expressed depending on other environmental fluctuations (temp etc).

Minor

1. Page 6 lines 174, 177 – Cannot start sentence with a number unless spelled out in full.

We thank each of our reviewers for the constructive and positive feedback on our manuscript. We have now addressed the reviewers' comments and each of our responses is noted below. Reviewer comments are in italics, and our responses follow, point by point. Line numbers refer to lines in the tracked changes version of our manuscript. The line numbers appear discontinuous in the tracked changes version of our manuscript because the tracked changes have not been accepted, please take note. Thank you very much.

Reviewer #1 (Remarks to the Author):

Q1. *McGrath et al. dives into the understudied realm of the biosynthetic potential of the Cariaco Basin including a focus on particle-associated vs. free-living lifestyles of microbes across euxinic and anoxic gradients. Their methods for analysis follow the trend set by several publications (Crits-Christoph et al. 2018, Nature; Nayfach et al. 2020 Nature Biotechnology; Paoli et al. 2022 Nature) for computational detection of biosynthetic gene clusters (BGCs) in MAGs here focusing on a somewhat unique habitat and microbial community composition relative to these previous studies. The overlay of metatranscriptomic data lends ecological relevance and we commend the authors on the inclusion of expression data in their study. However, the limited depth of their analysis and current unavailability of the code in a public repository (GitHub repo was empty as of 26.07.22) raises several concerns. See detailed comments below:*

R1. Thank you for these positive comments. We have resolved the issue of unavailability of the code and we have improved the annotation of our code. Our code is now available in the following GitHub repository: https://github.com/d-mcgrath/cariaco_basin. This is noted on line 1,234 (under Code availability section).

Q2. *Note to authors: continuous line numbers including the main text have been added so as to make the location of suggested edits more specific. A PDF with these line numbers is uploaded for reference. Please include line numbers in further versions of this manuscript.*

R2. Thank you for pointing out the omission of continuous line numbers. We retain those in our tracked changes revision.

Q3. *The use of the word “high-quality” for the MAGs is up for debate here. Please correct me if standards have changed however according to the minimum information about a metagenome-assembled genomes (MIMAG) Bowers et al. Nat. Biotechnol. 35, 725–731 (2017) I believe high quality should be greater than or equal to 90% mean completeness and less than or equal to 5% mean contamination. See methods of Paoli et al. 2022 Nature for further clarity on how to stratify MAGs by quality.*

R3. We have removed the term “high quality” from the manuscript as we agree that the cut-off we used, are not in accordance with the ones proposed by Bowers et al., 2017. The terminology “high-quality MAGs”, has now been replaced in our text with “MAGs”.

Q4. *One major item of concern: the Github repository https://github.com/d-mcgrath/cariaco_basin/ is empty. This is unacceptable, given the code availability statement in the manuscript lists all code are available. Please push well-commented code with a detailed README.*

R4. We apologize for this glitch in our github repository submission. We have fixed this. The code is now available along with a README file in the following GitHub repository: https://github.com/d-mcgrath/cariaco_basin.

Q5. *Line 79: “antimicrobial compounds (e.g., antibiotics, polyketides, RiPPs) – this sentence is somewhat misleading since many RiPPs and polyketides ARE antibiotics...perhaps NRPS would be more appropriate here?”*

R5. This is true, however, polyketides are also involved in other processes. They are described in anti-cancer, anti-cholesterol, and anti-inflammatory activities. We have modified this sentence starting on line 103 to say “...that can provide competitive advantages as antimicrobial compounds (NRPS, polyketides, RiPPs), or can”

Q6. *Throughout the manuscript: there are several locations e.g., lines 23-24, “remains virtually unexplored”, lines 68-69, lines 290-291 e.g., “for the first time in environmental surveys of BGCs, mapping of metatranscriptomes collected and preserved in situ to unveil...” where the authors claim to be “the first.” Given the recent plethora of publications analyzing metagenomes/MAGS (Nishimura et al. 2022 Scientific Data) and in some cases from open ocean environments along oxygen/depth gradients and particle-associated vs. free-living (e.g., Paoli et al. 2022 Nature), we recommend the authors please tone down their language. While these statements of novelty may have been true at the outset of the study it is certainly no longer the ‘first’ to conduct this type of analysis.*

R6. Our intention was not to claim that this is the first genome mining survey of BGCs. We were trying to highlight novel aspects of the work such as the investigation of BGCs in **anoxic and euxinic water columns**, as well as the use of RNA samples collected and stabilized **in situ** to quantify the expression of BGCs. We have clarified this point to make sure the “novelty” aspect is clearer in the introduction where we now say starting on line 98 “For this environmental survey of secondary metabolites, we use metatranscriptomes constructed from in situ filtration and preservation of water samples to compare the biosynthetic transcript expression profiles of particle-associated (PA > 2.7 μm) and free-living (FL; 0.2-2.7 μm) fractions. In situ filtration and fixation minimizes artifacts that can be introduced into RNA pools due to sample handling and physico-chemical changes²⁰.”

We also appreciate the reference to the Paoli paper (which was published after we had first drafted our manuscript) but the datasets used in their analyses include metagenomes only from Tara, Malaspina, HOTS and BATS, none of which (to our knowledge) is “truly” anoxic (undetectable oxygen) or euxinic.

We have removed the terminology “first” wherever it is inappropriate. Also, we have now included reference to the Paoli paper in the discussion starting on line 679 where we now say “Recently, an

analysis of >1000 publicly available marine metagenomes revealed ~40,000 putative BGCs⁴. Nonetheless, this analysis did not include anoxic/euxinic waters and did not incorporate information from transcripts.” Same applies when discussing the UMAP results (see the two paragraphs within that section starting on line 697).

Q7. *The authors highlight the oxycline as a key novel feature of their study. However, the analysis devoted to how BGC composition changes along the oxygen gradient is underdeveloped. For example, does oxygen availability have any impact on the number of BGC-associated or tailoring enzymes utilizing molecular oxygen (e.g., Rieske oxygenases) encoded within BGCs? Similarly, under euxinic conditions do protein-coding genes in BGCs exhibit altered ratios of sulfur-containing amino acids?*

R7. Thank you for this comment. We have followed this reviewer's suggestion and we have revised the text accordingly. It now reads (see starting on line 564): “We searched the BGCs for tailoring enzymes, including Rieske non-heme iron oxygenases (ROs). These enzymes contain oxygen-sensitive [2Fe-2S] clusters and are involved in synthesis of bioactive natural products⁶⁰. Overall, we detected 8 types of ROs in 32 MAGs encoding BGCs for terpenes, betalactones, T1PKS/T3PKS, phosphonates, RiPPs (lasso/thio/ranthipeptides, linear azole/azoline-containing peptides), NRPS (cyclodipeptides) and RiPP- and NRPS-like clusters. These ROs/ROs-domains were annotated to dioxygenases associated with degradation of aromatic amino acids (tyrosine/tryptophan), phosphonate and sulfur (taurine) cycling, pigment biosynthesis (carotenoids/betalain), and glyoxalase/bleomycin/validamycin dioxygenase superfamilies. This suggests that the identified ROs/ROs can be directly (e.g., synthesis of pigments, antibiotics) or indirectly (via nutrient/amino acid cycling) involved in the synthesis of these secondary metabolites.”

We also followed this reviewer suggestion and investigated whether BGCs would reveal altered ratios of sulfur-containing amino acids under euxinic conditions. We performed a Welch's two sample *t*-test on the ratio of sulfur-containing amino acids in BGCs prevalent in euxinic vs. oxic water column samples. Nonetheless, we did not find a significant difference ($P = 0.92$). Likewise, we tested for altered ratios of sulfur-containing amino acids in the entire genomes (MAGs) in order to include more “information” for the analysis (longer pieces of DNA). The *t*-test for the whole genome sulfur-containing amino acid ratio (for 9 abundant oxic/shallow anoxic MAGs with mean completeness/contamination of 88.5/1.01 and 13 abundant euxinic MAGs with mean completeness/contamination of 86.1/0.127) is statistically significant ($P = 0.02$). However, the mean sulfur ratio in euxinic genomes is lower (0.34) than the mean sulfur ratio for oxic/shallow genomes (0.37). Considering the caveats inherent when working with incomplete genomes any further discussion on these findings would require additional bioinformatical and statistical analyses which are beyond the scope of this paper. Also, given the insignificant result of the BGC test and the puzzling findings of this whole genome analysis, we don't feel comfortable to over-interpret these results using only hypothetical explanations. Other types of genomic approaches (e.g., SAGs that better capture representatives of the community) might be required to investigate this further.

Q8. *The discussion is remarkably short given the breadth of analysis that was carried out. Please consider amending with further discussion e.g., along the oxygen/sulfide gradients and other unique biogeochemical/taxonomic features of your sampling site.*

R8. The discussion has been expanded by the addition of text in response to several points raised by reviewers.

Q9. *Lines 116-117: How redundant were the BGCs detected? How fragmented? In general, the manuscript could use significantly more depth of analysis beyond a description, “which BGCs are there and expressed?” Rather than just reporting the total number of BGCs, comparative BiG-SLICE or BiG-SCAPE analysis would aid in the identification of similar and dissimilar biosynthetic gene cluster families across taxa and along the environmental gradients. Given that you have expression data, does analysis by BiG-MAP (Pascal Andreu et al., 2021) recover the same differentially expressed BGCs you identified using DESeq2?*

R9. Thank you for this suggestion. We tried applying the BiG-SCAPE analysis, but it showed that the biosynthetic gene cluster sequences are highly divergent/not redundant. Average family size of BiG-SCAPE clusters did not exceed 1 for any BGC class, meaning that most of the sequences were singletons that did not cluster with any other sequence. This could be due in part to the recovery of BGCs from a wide array of taxa in a very diverse environment. Therefore, given the complexity of our sample, the BiG-SCAPE output does not add much to our story. However, we have added a statement in our text starting on line 215 that states “BiG-SCAPE analysis revealed that the majority of detected BGCs longer than 10kb did not cluster together with the other 1,154 biosynthetic clusters. BiG-SCAPE created 15 gene cluster families (GCFs) of size 2, while 1,139 clusters were placed into singleton GCFs.” Use of BiG-SCAPE is also now described in the methods starting on line 1,142.

BiG-MAP was difficult to set up and run using the metatranscriptome setting. The BiG-MAP.map.py script crashed when input from the BiG-MAP.family.py script using the --metatranscriptome flag was used. Troubleshooting could not resolve the issue; we reached out to the developers but did not hear back. The tool only ran successfully when the --metatranscriptome flag was removed, but then analysis for e.g., housekeeping genes, was not carried out due to the tool’s assumption that it was mapping DNA-seq reads to the BGCs instead of RNA-seq reads. We don’t feel pursuing the application of BiG-MAP further is warranted.

Regarding fragmentation of the BGCs we recovered, 65% of them had a boundary on a contig edge, indicating potential incomplete recovery of the whole sequence for nearly two-thirds of predicted biosynthetic clusters. We have added a statement on line 213: “Sixty-five percent of our BGCs had a boundary on a contig edge, indicating potentially incomplete recovery of the whole sequence for nearly two-thirds of our predicted BGCs.” We also note in the methods on line 1,152 that “Gene clusters with a total length less than 10kb were discarded from downstream analysis to minimize the inclusion of fragmented BGCs in our data.”

Q10. *Furthermore, you could do dimensionality reduction (e.g., UMAP) using metagenomic GCF abundances and overlay sample metadata to learn more about biosynthetic structure along your gradients e.g., see the methods section of Paoli et al. 2022 Nature for an example of such UMAP analysis.*

R10. Thank you for this comment. We have followed this reviewer's suggestion, applied UMAP, and we have revised the text accordingly in both results and discussion sections. It now reads:

Starting line 362 of results: “We observed differences in BGC abundance and expression across size fraction in both the metagenomic and metatranscriptomic samples (Fig. 3a, 3b). Uniform Manifold Approximation and Projection (UMAP)⁴³ analysis of metagenomic and metatranscriptomic read recruitment to biosynthetic clusters primarily separated PA from FL sample types in most datasets (Fig. 3a, 3b). For the same size fraction and redox regime, UMAP analysis further separated most datasets between the two sampling points (May vs. November), particularly for BGC expression in oxycline and euxinic water features.”

Starting line 697 of discussion: “Analysis with UMAP of biosynthetic cluster abundances and expression profiles revealed a marked separation between the PA and FL size fractions in both the metagenomic and metatranscriptomic data. The niche preferences of taxa behind the MAGs we recovered, as well as the two different sampling times likely play a role in the observed differences in expression profiles of biosynthetic clusters in our PA vs. FL samples. We detected differences between sampling season and redox regime within the metagenomes. In the metagenomic samples, the PA euxinic and deep anoxic samples, as well as the FL euxinic samples clustered together (Fig. 3a). The abundance of metagenome reads mapped to MAGs across size fractions was similar at depths where oxygen is very limited or absent, and contributed to the clustering of BGC read abundances within these samples. The FL shallow anoxic and oxycline, and the PA oxycline samples formed three distinct clusters, suggesting differing redox conditions shaped BGC composition and abundance in these samples. Within oxycline samples, the influence of oxygen and the separation between PA and FL size fractions is evident.

UMAP analysis of the metatranscriptomic data revealed BGC expression profiles that differentiated primarily by size fraction as well as season of sampling (Fig 3b). Some overlap is observed between BGCs expressed in the PA and FL fractions, consistent with the idea that some taxa may transiently associate with particles as they sink through the water column. Paoli et al. (2022)⁴ examined MAGs recovered from PA and FL fractions in global datasets (that did not include ODWCs) and they found genes for terpenes and RiPPs enriched in the FL fraction, and NRPS and PKS genes enriched in PA samples. This supports the idea that taxa and the genes they carry are shaped by their FL vs. PA lifestyle (niche requirements). Seasonal differences in primary productivity can also shape microbial communities and the genes they express. In Cariaco Basin, upwelling of nutrients occurs between January and March, fueling increased primary productivity⁷². This may be a contributing factor to the observed separation of most PA vs. FL BGC profiles (Fig. 3a, b) because in the FL state microorganisms will experience environmental shifts more directly than those protected within particles.”

If this reviewer prefers, we can add highlighted ovals around the clusters mentioned in the UMAP figure, as was done by Paoli et al. 2022.

Q11. Lines 200-201, “Clusters of remaining MAGs classified by antiSMASH to encode ladderanes may synthesize other products, including polyunsaturated hydrocarbons.” Can you quantify how many of the ladderane clusters have genes characteristic of PUFA clusters? PUFAs will have very distinct marker genes e.g., *pfaA*. I also believe they are more frequently classified as polyketide synthases than as ladderanes. Please verify the accuracy of this statement and expand.

R11. Thank you for this useful suggestion. We have checked for the *pfaABCDE* operon involved in production PUFAs in marine bacteria. Our analysis didn’t reveal misclassified *pfa* genes in the

ladderane BGCs. To avoid misinterpretation, we have rephrased the text starting on line 435, and now it reads: “Clusters of remaining MAGs encoding ladderanes may serve unknown functions in Cariaco Basin. Plausible *in silico* explanations for ladderanes in non-anammox taxa include possible involvement in fatty acid biosynthesis⁴⁴ and in lineage divergence of closely related taxa via acquisition of ladderane genes⁴⁶. These could apply to the Cariaco Basin but needs to be validated experimentally.”

Q12. *Line 220-221, the discussion of redox-active metabolites is lacking since redox-active do much more than oxidative stress for example, iron or phosphorus cycling. Please discuss, in the context of McRose et al. 2021 Science and related work*

R12. Thank you for this comment. We have added the following text on starting on line 460: “Nevertheless, expression of phenazines could increase microbial fitness in Cariaco Basin by enhancing phosphorus cycling. Within the redoxcline of the Cariaco Basin exists a challenging variability in phosphate concentrations whose fate (precipitation vs. remobilization) is controlled by the delivery of iron and manganese in the water column⁵³. Phenazines are phosphorus/iron-regulated antibiotics suggested to promote microbial growth under phosphorus starvation via solubilization of phosphates through reduction of iron oxides⁵⁴.”

Q13. *Lines 228-229 - Is there a reason you chose this more ‘custom approach’ for detection of antibiotic resistance genes in BGCs compared to using e.g., ARTS (Mungan et al. 2020 Nucleic Acids Research). How do your results compare to accepted tools for this analysis?*

R13. Thank you for this suggestion. We did run ARTS, and this analysis revealed that we hadn’t used a strict enough e-value for our previous analysis that led to overestimation of the antibiotic resistance genes in the BGCs. We have now restricted our discussion to genes identified by ARTS, and have revised and shortened the section on antibiotic resistance.

Line 218 of results now says: “Antibiotic Resistance Target Seeker^{33,34} identified putative antibiotic resistance genes within some BGCs.”

Line 544 of results now reads “We applied Antibiotic Resistant Target Seeker (ARTS^{33,34}) to detect antibiotic resistance genes in our BGCs. We detected only 4 types of proteins/protein domains involved in resistance (Supplementary Table 5). These include 13 MAGs that had ABC and RND efflux pumps, RND-type membrane proteins of the efflux complex MexW/MexI/MexH⁵⁶, and 8 MAGs that contained pentapeptide repeats⁵⁷. These were all associated with BGCs that coded for terpenes, bacteriocins, T1PKS/T3PKS, homoserine lactone (hserlactone), NRPS/NRPS-like, betalactones, arylpolyenes and hgIE-KS”

A table of the results of the ARTS analysis is presented as Supplementary Table 5.

In the discussion on line 812 we added the statement “Co-localized genes encoding antibiotic resistance were present in the BGCs we identified.”

Q14. *Lines 309: “While it is less clear how free-living microbes could benefit from release of*

secondary metabolites...” - please discuss further here- some possible explanations and examples in the literature e.g., nutrient acquisition

R14. Thank you for this suggestion. We have revised the text accordingly and added discussion as suggested by this reviewer. Now it reads (starting line 801): “The role of secondary metabolites in microbial fitness is an open debate because possession of secondary metabolism can enhance the overall fitness, but not all products of secondary metabolism will necessarily have an effect on the producer⁷³. Nonetheless, secondary metabolites are reported to affect niche utilization, shape microbial community assembly, and act as a functional trait driving ecological diversification among closely-related bacteria inhabiting the same microenvironments^{74,75}. Likewise, the example of phenazines and phosphorus acquisition can be a paradigm of dual/pleiotropic functions of secondary metabolites where they can serve as potential antibiotics and regulators of nutrient cycling.”

Q15. *Finally, although I recognize this is not trivial and possibly not feasible given expense and time constraints, some form of experimental characterization of one or more BGCs or producing organism would be a valuable addition to validate computational predictions.*

R15. We agree this is a good next step, but this is outside the scope of the current project.

Reviewer #2 (Remarks to the Author):

This is an interesting study on BGC prevalence in permanently anoxic waters, and one of few studies to go beyond BGC potential and look at expression. Some comments below would be good for the authors to clarify to help improve the understanding of their study.

Q1. *Page 3 line 70 – It was not completely clear to me why the roles of particles in the production of BGCs was a critical gap in knowledge. It may well be but thinks this needs to be better justified and explained in the text early on as to why it so important to study particles in relation to BGCs. How are particles important in this anoxic, sulfidic zone? Are they offering protection, a niche where metabolites can be concentrated and exchanged?*

R1. Thank you for the positive feedback. This sentence has now been revised at line 88 to clarify it: “Further, analyses of size-fractionated water samples is required in order to assess the role of particles in the production of secondary metabolites in the environment. Particles provide colonizable, nutrient-rich substrates where metabolites can be concentrated and exchanged and can provide protection for oxygen- or sulfide-sensitive microbiota.”

Q2. *Page 3 line 71 – Perhaps instead of ‘a large number’ which is very subjective, use ‘565’ or ‘over 500’ to add some level of detail here.*

R2. We have corrected this to indicate the number as suggested.

Q3. Page 3 line 76 – PA and FL need to be defined here and as per above the significance of analysing particles made clearer.

R3. We have defined these terms at first use where we now say “particle-associated (PA > 2.7 µm) and free-living (FL; 0.2-2.7 µm) fractions.”

Q4. Page 3 line 83 – As per the established definition of MAG quality by the MIMAG (Minimum information about a metagenome-assembled genome) from Bowers et al (2017) I was under the impression that high quality MAGs needed to be closer to 90% bin completion. Could the authors clarify?

R4. Thank you for this comment. We addressed this above.

Q5. Page 4 line 102 – What is the definition of ‘enriched’ in this context? Enriched from what? Are they at higher numbers, abundances compared to others and if so what is the criteria and what values can be put to this if any?

R5. On this page “enriched” has been replaced with “more abundant” to improve clarity.

Q6. Page 4 lines 116-118- In addition to terpenes, was there any evidence for siderophores, ectoines, or homoserine lactones production/capacity?

R6. Thank you for this suggestion. We have added to this sentence starting on line 184 “...as well as four ectoine clusters...”

Q7. Page 5 line 132 – Re intercellular communication, as per the query above were any AHL-based (or other) signalling systems observed?

R7. We thank the reviewer for this comment. We have not detected AHL-based signaling systems/pathways in our dataset. To our knowledge these are more prominent in quorum sensing related to marine pathogenicity and/or colonization of higher eukaryotic hosts (e.g., sponges, shrimps, alga; e.g., Kjelleberg et al., 1997; Reen et al., 2019 and references therein). We have detected various lactones, terpenes and non-ribosomal peptides that may play a role in signaling. Also, we identified betalactones with ARTs, suggested to have quorum quenching activities (please see previous response to reviewer 1 where we describe the antibiotic resistance genes that were detected with ARTs). We revised the text on line 252 as follows: “The transcripts were predominantly from terpene, non-ribosomal peptide, and lactone clusters with inferred antibiotic activity, as well as roles in oxidative stress and cell-to-cell signaling.” Please see references below:

Kjelleberg et al., (1997). Do marine natural products interfere with prokaryotic AHL regulatory systems? *Aquat Microb Ecol.* **13**:85-93.

Reen et al., 2019. Quorum Sensing signaling alters virulence potential and population dynamics in complex microbiome-host interactomes. *Front Microbiol.* **10**:2131.

Q8. Page 6 lines 163-164 – In terms of the overlap of BGCs between PA and FL fractions would

it not be expected that at least some taxa would occupy both niches? I am not an expert in particle formation but would not free-living marine microbes be the source of those residing on/in particles?

R8. Yes, we agree with the reviewer that it is logical to think that PA microbes would come on and off particles as they sink, so that some overlap in these two fractions (FL and PA) would be expected. Therefore, the observed differences would likely result from microbes that colonize particles in the upper water column and remain inside the particles, protected, as it sinks into zones where conditions are no longer ideal for them to be found in the FL state. We have expanded on this theme at this location of the paper by stating on lines 283: “It is also possible some MAGs enriched in the FL fraction dissociated from particles during sample processing. Microorganisms likely also attach to, and disassociate from, particles as they sink through the water column. Some cells that associate with particles in the surface ocean may remain trapped within particles as they sink into realms that no longer favor their survival in the FL state. These hypotheses require further investigation, as do other possible roles these secondary metabolites might play in the free-living state, including grazer avoidance.”

Q9. *Page 7 line 198 - Can the authors postulate on any potential relevance to annamox at shallow oxycline depths?*

R9. Signatures of anammoxers were actually prevalent in the shallow anoxic depths (where oxygen is undetectable). Although we haven't tried to quantify anammox in Cariaco, previous research on open ocean oxygen depleted systems are in accordance with our findings. Dalsgaard et al. 2012, Thamdrup et al. 2021 and several others have shown that anammox (and denitrification) were almost exclusively recorded when the in situ O₂ concentration was below detection, indicating that the induction of these processes is highly oxygen sensitive. Please see references below.

Dalsgaard et al. (2012). Anammox and denitrification in the oxygen minimum zone of the eastern South Pacific. *Limnol. Oceanogr.* **57**(5): 1331-1346.

Thamdrup et al. (2021). Anammox bacteria drive fixed nitrogen loss in hadal trench sediments. *PNAS* **118** (46) e2104529118.

Q10. *Page 7 lines 205-215 – Do microbial communities associated with particles form biofilm-like phenotypes? If so there would be expected increased oxidative stress within the biofilm thus was there an increased prevalence of BGCs related to oxidative stress in particles (compared to free living)?*

R10. Yes, generally they do form biofilms. However, in our study we have not searched either microscopically or genetically for the detection of biofilm-like phenotypes. Yet, some specific secondary metabolites we see have been suggested to play a role in biofilm formation, such as phenazines and RiPPs (e.g., thiopeptides). Please see sentence added on line 458 that now says “Phenazines are redox-active compounds known to contribute to formation of bacterial biofilms and to cause debilitating oxidative stress in targeted cells by forming intracellular free radicals of both reactive oxygen and nitrogen species^{51,52}”. Likewise, regarding RiPPs, on line 103 we now mention the possible role of RiPPs and phenazines in biofilm formation.

Q11. *Page 22 line 443- I noted in the sampling that different samples were taken at different seasons. There was no real analysis and discussion on this and I am curious as to whether BGCs could be differentially expressed depending on other environmental fluctuations (temp etc).*

R11. Thank you for this suggestion. We performed the UMAP analysis (see answers above and the discussion). In any case, for this study we only sampled twice and while we saw some differences, it would be an over-interpretation of the data to make further claims about seasonality with only two timepoints.

Minor

Q12. *Page 6 lines 174, 177 – Cannot start sentence with a number unless spelled out in full.*

R12. Thank you, this has been corrected.

Reviewer #1 (Remarks to the Author):

McGrath et al. have done significant work to respond to reviewers requests. I have a few minor requests below with respect to the correctness of the wording in select cases, but overall applaud the authors on their thorough analysis.

-Lines 251-253

"...exists a challenging variability in phosphate concentrations whose fate (precipitation vs. remobilization) is controlled by the delivery of iron and manganese in the water column⁵³. Phenazines are phosphorus/iron-regulated antibiotics suggested to promote microbial growth under phosphorus starvation via solubilization of phosphates through reduction of iron oxides⁵⁴. Expression of genes associated with redox-cycling antibiotics was found primarily in FL metatranscriptomes at all water layers."

We thank the author for considering alternative roles for the phenazines in light of recent work in this field. Did the authors analyze the phosphate, iron, and manganese concentrations in the water columns and whether there was any association with phenazine BGC expression? The statements made are descriptive while the study would benefit from a more quantitative analysis.

-Line 308: Just a cosmetic detail: "secondary metabolites synthesized by the recovered MAGs."

The MAGs themselves do not produce the secondary metabolites, rather the organisms. Consider rewording.

-Lines 320 – 322: "Nonetheless, this analysis did not include anoxic/euxinic waters and did not incorporate information from transcripts."

This statement is unfortunately incorrect as worded. The TARA dataset described by Paoli et al. does cover some oxygen-depleted zones particularly in the Arabian Sea (see <https://www.microbiomics.io/ocean/map/>) and based on metadata includes >20 metagenomes samples from sites with oxygen levels measured at less than 10 $\mu\text{mol/kg}$. Moreover, the study by Paoli et al. does incorporate information from metatranscriptomic data for which a detailed study of BGC expression was examined for selected MAGs (e.g., see Figure 3c and Extended Data Figure 7). The metatranscriptomic data were also previously extensively analyzed by Salazar et al. 2019: <https://pubmed.ncbi.nlm.nih.gov/31730850/>. Please revise the wording of these lines for correctness.

-Lines 773 – 775: "Paoli et al. (2022)⁴ examined MAGs recovered from PA and FL fractions in global datasets (that did not include ODWCs)..."

See above comment on lines 320-322. It's clear that ODWCs represent a knowledge gap not specifically focused on by Paoli et al., but these global datasets do overlap therefore "did not include" is strictly incorrect. We again encourage the authors to revise the wording and emphasize the knowledge gap rather than take a defensive tone against other studies of this kind.

Reviewer #2 (Remarks to the Author):

The authors appear to have taken into account comments and suggestions I have made and think this has improved their manuscript to facilitate publication.

REVIEWERS' COMMENTS

Reviewer #1 (Remarks to the Author):

McGrath et al. have done significant work to respond to reviewers requests. I have a few minor requests below with respect to the correctness of the wording in select cases, but overall applaud the authors on their thorough analysis.

Thank you.

-Lines 251-253

“...exists a challenging variability in phosphate concentrations whose fate (precipitation vs. remobilization) is controlled by the delivery of iron and manganese in the water column⁵³. Phenazines are phosphorus/iron-regulated antibiotics suggested to promote microbial growth under phosphorus starvation via solubilization of phosphates through reduction of iron oxides⁵⁴. Expression of genes associated with redox-cycling antibiotics was found primarily in FL metatranscriptomes at all water layers.”

We thank the author for considering alternative roles for the phenazines in light of recent work in this field. Did the authors analyze the phosphate, iron, and manganese concentrations in the water columns and whether there was any association with phenazine BGC expression? The statements made are descriptive while the study would benefit from a more quantitative analysis.

Thank you. The cruises conducted in Cariaco Basin in 2014 were primarily microbiology surveys aiming to describe the prokaryotic communities in the water column of Cariaco Basin (e.g., collecting samples for cell counts, hybridization experiments, amplicons, metagenome, and metatranscriptome analyses). For this reason, samples for nutrients (with the exceptions of NO_x and NH₄) and micronutrients analyses were not collected. The statements that we made are based on observations coming from McParland et al., 2015. That paper measured and speciated phosphorus, and suggested that the overall biogeochemical cycling of P may be altered in unexpected ways due the presence of iron and manganese in the water column of Cariaco Basin. We agree that it would be beneficial to have concentrations of those compounds in order to look for associations with phenazines. Nonetheless, this is not feasible with the current datasets we have.

-Line 308: Just a cosmetic detail: “secondary metabolites synthesized by the recovered MAGs.”

The MAGs themselves do not produce the secondary metabolites, rather the organisms. Consider rewording.

Thank you for catching this. We have rephrased and now reads: “The presence of various biosynthetically important protein domains present in the recovered BGCs suggests a variety of diverse chemical transformations and post-translational modifications that could shape the secondary metabolites synthesized by the prokaryotes identified in the water column of Cariaco Basin.”

-Lines 320 – 322: “Nonetheless, this analysis did not include anoxic/euxinic waters and did not incorporate information from transcripts.”

This statement is unfortunately incorrect as worded. The TARA dataset described by Paoli et al. does cover some oxygen-depleted zones particularly in the Arabian Sea (see <https://www.microbiomics.io/ocean/map/>) and based on metadata includes >20 metagenomes samples from sites with oxygen levels measured at less than 10 $\mu\text{mol/kg}$. Moreover, the study by Paoli et al. does incorporate information from metatranscriptomic data for which a detailed study of BGC expression was examined for selected MAGs (e.g., see Figure 3c and Extended Data Figure 7). The metatranscriptomic data were also previously extensively analyzed by Salazar et al. 2019: <https://pubmed.ncbi.nlm.nih.gov/31730850/>. Please revise the wording of these lines for correctness.

*Yes, we inspected the metagenomes list used for the Paoli paper. Although, as pointed out by the reviewer, while **low** oxygen samples from Arabian Sea were included in the analysis, “true” anoxia cannot be easily detected. Most of the commonly used ways to measure oxygen have a detection limit of a few nM. The presence of sulfide ensures anoxic conditions because this compound is quickly oxidized in the presence of oxygen. To avoid further confusion, we rephrased the statement to read “this analysis did not include samples from sulfidic waters”.*

-Lines 773 – 775: “Paoli et al. (2022)⁴ examined MAGs recovered from PA and FL fractions in global datasets (that did not include ODWCs)...”

See above comment on lines 320-322. It’s clear that ODWCs represent a knowledge gap not specifically focused on by Paoli et al., but these global datasets do overlap therefore “did not include” is strictly incorrect. We again encourage the authors to revise the wording and emphasize the knowledge gap rather than take a defensive tone against other studies of this kind.

We rephrased this to “examined MAGs recovered from PA and FL fractions in global datasets (that did not include sulfidic end-members)” to avoid confusion.

Reviewer #2 (Remarks to the Author):

The authors appear to have taken into account comments and suggestions I have made and think this has improved their manuscript to facilitate publication.

Thank you.